# Association between Soft-Drink Intake and Obesity, Depression, and Subjective Health Status of Male and Female Adults

**DOI:** 10.3390/ijerph181910415

**Published:** 2021-10-03

**Authors:** Ji-Myung Kim, EunJung Lee

**Affiliations:** 1Food and Nutrition Major, Division of Food Science & Culinary Arts, Shinhan University, Uijeongbu-si 11644, Gyeonggi-do, Korea; kjm@shinhan.ac.kr; 2Food and Nutrition Major, School of Wellness Industry Convergence, Hankyong National University, Anseong-si 17579, Gyeonggi-do, Korea

**Keywords:** soft drinks, obesity, depression, subjective health status, male and female adults

## Abstract

This study explores the association between soft-drink intake and obesity, depression, and subjective health status in male and female adults. Soft-drink consumption has become a serious public health issue worldwide. Participants of this study were selected (*n* = 3086) from the respondents in the Seventh Korea National Health and Nutrition Examination Survey 2016 and divided into the non-soft-drink group (men, 502; women, 1117) and the soft-drink group (men, 684; women, 783). Soft-drink intake data were obtained, using a validated quantitative food frequency questionnaire and the 24 h dietary recall. Total energy intake and fat density were significantly higher, but the nutrient densities of carbohydrate, total dietary fiber, and micronutrients were significantly lower in the soft-drink group than in the non-soft-drink group. In men, soft-drink intake was found to be significantly associated with a lower risk of depression. Conversely, in women, soft-drink intake was found to be significantly associated with the higher risks of depression, obesity, and poor subjective health status. Therefore, less frequent soft-drink intake may prevent obesity and depression in Korean women.

## 1. Introduction

Soft drinks mainly include soda water, syrup, or other carbonated or noncarbonated beverages containing substantial amounts of added sugars [1]. The excessive consumption of soft drinks has become a serious public health issue worldwide [2,3], and it has been suggested that a high consumption of sugar-sweetened beverages may contribute to weight gain and obesity by increasing overall energy intake [4]. Studies have shown that depression in obese people is more than twice as high as that in people with normal weight, and that there is an association between depression and obesity [5]. Depression is associated with reduced intake of fruit and vegetable [6], and its severity is related to higher total energy intake from saturated fats and sugars [7] and to higher consumption of sweet foods [8]. Recently, there has been a lot of research on the effects of soft drinks on mental health. A meta-analysis has shown that higher consumption of sugar-sweetened beverages is positively associated with depression [9]. 

Studies from Europe and the US have also shown a consistent link between soft-drink consumption and mental health problems among adolescents and older adults [10,11]. In recent years, studies have reported that higher levels of soft-drink consumption in Chinese adults and adolescents are associated with increased symptoms of depression [12]. Furthermore, a study demonstrated that frequent consumption of soft drinks is linked to a higher likelihood of depressive symptoms among workers in Japan [13]. In Korea, study results have indicated that unhealthy dietary habits, such as drinking soft drinks, have relatively detrimental effects on perceived physical and mental health among Korean adolescents [14]; however, as far as we know, no research has been conducted in this regard among Korean adults.

The results indicate that discouraged dietary habits have relatively detrimental effects on perceived physical and mental health in Korean adolescents. With regard to mental health, in instances of depressive experiences, it was found that the higher the level of stress, the lower was the subjective health status perception [15]. Because subjective health status can be used as an indicator to predict a person’s health status, it has been widely used as a health-related indicator [16], and it has reported that there is a correlation between subjective health status and depression [17]. However, the correlation between soft drink intakes and subjective health status perception has not yet been confirmed. 

The prevalence of obesity or depression differs by sex, with higher prevalence of obesity in men in developed countries [18] and of depression in women [19]. Therefore, there may be sex differences in the effects of soft-drink intake on obesity and depression. For this reason, this study aimed to identify the association between soft-drink intake and obesity, depression, and subjective health status according to sex in adults based on the 2016 Korea National Health and Nutrition Examination Survey (KNHANES VII-1).

## 2. Materials and Methods

### 2.1. Study Design and Participants

This study was based on the data from the 2016 KNHANES [20]. KNHANES is a nationwide survey using a stratified multistage probability sampling design and consists of health interviews, behavioral and nutrition surveys, and medical examinations. A total of 8150 individuals completed health and behavior interviews, health examinations, and nutrition surveys from January to December 2016. Prospective participants who fall under one of the following were excluded from the study:(1)under 18 or over 65 years of age (*n* = 3400),(2)without dietary intake records or food frequency questionnaire (FFQ) data (*n* = 1379),(3)with implausible daily energy intake (<500 or ≥5000 kcal/day) (*n* = 26), missing values for depression (*n* = 220), or missing values for sociodemographic variables (*n* = 39).

Subsequently, 3086 participants (1186 men and 1900 women) were included in this study. The 2016 KNHANES was exempted from research ethics approval from the Bioethics and Safety Act of the Institutional Review Board of the Korea Centers for Disease Control and Prevention. All participants provided written informed consent.

### 2.2. General Measurements

Sociodemographic factors, such as age, sex, occupation, education, income, marital status, and residence, were collected using a self-reported questionnaire. Smoking status, alcohol consumption, and physical activity data were collected using health and behavior interviews. Education levels were categorized in accordance with their graduation status (elementary school, middle school, or high school or higher). Marital status was categorized as either married or unmarried, and the household income was categorized as low, middle–low, middle–high, and high. Place of living was classified into two groups: urban and rural. Smoking status was divided into yes (current smoker) or no (former/never smoker), and alcohol intake was divided into yes (current drinker) or no (nondrinker). Physical activity was also divided into two categories: yes (practice physical activity of mid intensity over 2.5 h per week or of high intensity or mid and high mixed intensity over 1.25 h per week (1 min of high-intensity physical activity is equal to 2 min of mid-intensity physical activity)) and no (no exercise) according to the KNHANES’s guideline by the KCDC [21,22].

### 2.3. Health Behavior Measures

#### 2.3.1. Obesity

Anthropometric data were collected by a team of experts by using a consistent, standardized methodology during the health examinations. Height was measured to the nearest 0.1 cm using stadiometer (Seca 225, Seca., Hamburg, Germany). Waist circumference was measured using a measuring tape (Seca 200; Seca, Hamburg, Germany) to the nearest 0.1 cm. Similiarly, weight was measured to the nearest 0.1 kg in light clothing, with shoes off, using a calibrated balance scale (GL-6000-20, G-tech, Korea). To maintain the accuracy of the measurements, the instruments are replaced with newly calibrated devices every year. Body mass index (BMI) was calculated as weight (kg) divided by height squared (m^2^), and the participants were classified into three categories based on their BMIs (kg/m^2^): underweight (BMI < 18.5), normal weight (18.5 ≤ BMI < 25), or obesity (BMI ≥ 25), according to the KNHANES’ guideline by the KCDC [21]. The waist-to-height ratio (WHtR) was calculated by dividing waist by height for measuring central adiposity [23]. Relative fat mass (RFM) was calculated using height and waist circumference according to sex in order to estimate total body fat percentage. The RFM formula is as follows [24]: men: 64 − (20 × height/waist circumference); women: 76 − (20 × height/waist circumference).

#### 2.3.2. Subjective Health Status

Subjective health status data were measured using one item in a health and behavior interview in the KNHANES 2016 [20]. Participants were asked, “In general, how would you rate your health?” and they answered on a five-point scale of 1 (very good), 2 (good), 3 (moderate), 4 (bad), or 5 (very bad). In this study, subjective health status was classified into two categories: good (self-rating of very good, good, or moderate) or bad (self-rating of bad or very bad) [25].

#### 2.3.3. Depression

Participants with depression were identified on the basis of their answers to the Patient Health Questionnaire (PHQ)-9 section of the health survey in KNHANES 2016 [20]. Participants were asked, “How often have you been bothered by any of the following symptoms over the last two weeks?” The PHQ-9 uses nine items to measure the severity of depressive symptoms. Each of the nine items was rated on a four-point scale of 0 (not at all), 1 (several days), 2 (more than half the days), and 3 (every day), and the answers were summed to obtain the total PHQ-9 score. In line with the methods used in previous studies, in this study, participants with a PHQ-9 score ≥10 (27 points) were defined as having depression [26].

### 2.4. Dietary Intake Measures

#### 2.4.1. Dietary Intake

Dietary intake data were obtained using a 1 day 24 h dietary recall protocol [27] as part of the nutrition survey. The dietary data were collected by trained dietitians at the participant’s home, 1 week after the completion of health interview and health examination surveys. Detailed descriptions of the type and amount of all foods consumed on the day before the interview, including snacks, beverages, and ingredient information for all sauces and condiments were requested. Additionally, information regarding the time and the place of each meal was recorded. The 24 h dietary recall began with the first meal or beverage consumed at waking until midnight of the reporting day. Trained dietitians administered the 24 h dietary recall using measuring cups, portion-size booklets, and photographs. All completed records were checked by a research dietitian for accuracy. Using the 24 h dietary recall, total energy and nutrient intake were calculated on the basis of the food composition table published by the Rural Development Administration of Korea [28]. Macronutrient intake was evaluated as a percentage of the total energy intake (% kcal), and nutrient density (nutrient amount per 1000 kcal) was calculated. 

#### 2.4.2. Soft-Drink Intake

Soft-drink intake data were obtained using two dietary intake survey methods: (1) validated quantitative food frequency questionnaire (FFQ), and (2) 24 h dietary recall. The FFQ was developed on the basis of dietary intake data from the KNHANES and was previously validated using dietary records taken over 3 days during each of the four seasons in 124 participants, as described elsewhere [29]. FFQ requested information on the participants’ consumption of 112 food items during the past year. The participants’ food intake frequency was quantified according to nine categories: “never or seldom”, “once a month”, “two to three times a month”, “one to two times a week”, “three to four times a week”, “five to six times a week”, “once a day”, “twice a day”, and “three times or more every day”. The portions were categorized as one of the three sizes depending on the type of food consumed: 0.5, 1.0, and 1.5 portions, as presented in the KNHANES’ guideline by the KCDC [20,30].

Soft-drink intake was defined as carbonated drinks (i.e., cola, cider, and other soft-drink beverages) in the questionnaire in this study. First, soft-drink intake by FFQ was analyzed in the following manner: for the analysis of soft drink beverage consumption, this study utilized the intake frequency of soft drinks, serving size (1.0 portion = 1 cup, 200 mL), which was converted into the weekly consumption frequency of soft drinks according to the FFQ data. Daily soft-drink intake (mL) was also calculated using the soft-drink intake frequency data and serving size by FFQ.

Second, the soft-drink intake by 24 h dietary recall was analyzed by summing and calculating per day consumption (mL) of soft drinks including cola, cider, and other soft-drink beverages in accordance with the soft-drink classification by FFQ.

### 2.5. Statistical Analysis

All the statistical analyses were performed using SAS software (version 9.4; SAS Institute, Cary, NC, USA). To avoid biased results, the comprehensive survey design consisted of multistage, stratified, and clustered samples and survey weights which reflect the estimates of the entire Korean population. The participants were categorized into two groups on the basis of the soft-drink consumption frequency obtained from the FFQ data: (1) non-soft-drink group (never or seldom) and (2) soft-drink group (greater than once a month). The analyses were conducted separately for both men and women. Total energy and nutrient intakes were expressed as means ± standard errors. The categorical variables, including age, education, income, marital status, alcohol intake, smoking, physical activity, subjective health status, BMI, and depression categories, were expressed as frequency and percentages to compare between the non-soft-drink group and the soft-drink group, using the chi-square test. Nutrient intake by soft-drink consumption was analyzed using a general linear model. The odds ratios (ORs) and the 95% confidence intervals (CIs) were calculated, using logistic regression analyses. The prevalence of obesity, bad subjective health status, and depression were compared with soft-drink consumption, using multiple logistic regression analyses before (Model 1) and after adjusting variables (Model 2). The variables included age, household income level, education level, marital status, occupation, physical activity, BMI, and total energy intake for men, and age, education level, marital status, level of alcohol consumption, BMI, and total energy intake for women. The statistical significance was set at *p* < 0.05.

## 3. Results

### 3.1. General Characteristics and Soft-Drink Intakes of Participants

Table 1 shows the sociodemographic characteristics and soft-drink intake by the participants according to the frequency of soft-drink intake by sex. Among the participants, 62.3% (*n* = 684) of men and 44.3% (*n* = 783) of women consumed soft drinks more than once a month. The mean age was significantly lower in the soft-drink group than in the non-soft-drink group (*p* < 0.001 for men; for women), and the proportion of the population in their 20 s was significantly higher in the soft-drink group than in the non-soft-drink group (*p* < 0.001 for men and women). The soft-drink group had a significantly higher rate for the variable education and marriage compared with the non-soft-drink group (*p* < 0.001 for men and women). The proportion for the variable occupation was significantly lower in the soft-drink group compared with the non-soft-drink group for men (*p* = 0.029), but not for women. The proportion for the variable alcohol intake was significantly higher in the soft-drink group compared with the non-soft-drink group for women (*p* < 0.001), but not for men. The soft-drink group had significantly lower WHtR and RFM than the non-soft-drink group (*p* < 0.001 for men and women).

The weekly soft-drink frequency in the soft-drink group was 2.0 ± 0.1 for men and 1.5 ± 0.1 for women. The daily soft-drink intake obtained from the FFQ method in the soft-drink group was 71.3 ± 3.5 mL for men and 46.5 ± 3.1 mL for women. The proportion of drinking soft drinks analyzed by the 24 h recall method was significantly higher in the soft-drink group than in the non-soft-drink group (*p* < 0.001; 32.5% vs. 8.9% in men; 28.3% vs. 8.3% in women, respectively). The soft-drink intake analyzed by the 24 h recall method was significantly higher in the soft-drink group than in the non-soft-drink group (*p* < 0.001 both for men and women). 

### 3.2. Health-Related Outcomes of Participants

Table 2 shows the participants’ health-related outcomes, such as obesity, subjective health status, and depression according to the frequency of soft-drink intake by sex; there were no significant differences in the health-related outcomes in men, but there were significant differences in obesity in women. For women, the proportion of underweight was significantly higher in the soft-drink group than in the non-soft-drink group (*p* = 0.001).

### 3.3. Daily Nutrient Intake of Participants

Table 3 shows the participants’ daily nutrient intake. The energy intake in the soft-drink group was significantly higher than in the non-soft-drink group (*p* < 0.001 for men and women). With respect to the distribution of energy intake of macronutrients, the soft-drink group had a significantly lower carbohydrate intake (*p* < 0.001 for men and women) and a higher protein (*p* < 0.001 for only men) and fat intake (*p* < 0.001 for men and women) than the non-soft-drink group. The nutrient densities of proteins (*p* < 0.001 for only men) and fats (*p* < 0.001 for men and women) in the soft-drink group was significantly higher than those in the non-soft-drink group; however, the nutrient densities of carbohydrate (*p* < 0.001 for men and women), total dietary fiber (*p* < 0.001 for men and women), calcium (*p* < 0.001 for men and women), phosphorus (*p* < 0.001 only for women), iron (*p* = 0.025, for men; *p* < 0.001 for women), potassium (*p* < 0.001 for men and women), thiamin (*p* = 0.013 for men; *p* < 0.001 for women), riboflavin (*p* < 0.001 for women), and vitamin C (*p* < 0.001 for men and women) were significantly lower than those in the non-soft-drink group.

### 3.4. Relationship between Soft-Drink Intake and Health-Related Outcomes

Table 4 presents the ORs for health-related outcomes based on the soft-drink intake. In men, soft drink intake in daily increments of 10 mL by 24 h recall was found to be significantly associated with a lower prevalence of depression (Model 1: OR, 0.957; 95% CI, 0.918–0.998; *p* < 0.05; Model 2: OR, 0.950; 95% CI, 0.906–0.996; *p* < 0.05; Model 3: OR, 0.953; 95% CI, 0.911–0.997, *p* < 0.05). 

For women, with respect to obesity, there was no significant association with obesity prevalence in Model 1. However, with the adjustments for Models 2, 3, and 4, the soft-drink group (Model 2: OR, 1.658; 95% CI, 1.237–2.223, *p* < 0.001: Model 3: OR, 1.658; 95% CI, 1.237–2.221, *p* < 0.001: Model 4: OR, 1.593; 95% CI, 1.183–2.146; *p* < 0.01) showed a significant association with higher prevalence of obesity compared with the non-soft-drink group, and the soft-drink intake in daily increments of 10 mL according to the FFQ (Model 2: OR, 1.040, 95% CI, 1.016–1.066, *p* < 0.01: Model 3: OR, 1.036; 95% CI, 1.010–1.063, *p* < 0.01: Model 4: OR, 1.034; 95% CI, 1.010–1.058; *p* < 0.01) and according to 24 h recall (Model 2: OR, 1.013; 95% CI, 1.004–1.022, *p* < 0.01: Model 3: OR, 1.012; 95% CI, 1.003–1.021, *p* < 0.01: Model 4: OR, 1.011; 95% CI, 1.002–1.020; *p* < 0.05) was found to have a significant association with higher prevalence of obesity.

With respect to subjective bad health status, soft-drink intake in daily increments of 10 mL according to the FFQ (Model 1: OR, 1.042; 95% CI, 1.016–1.070; *p* < 0.01; Model 2: OR, 1.062; 95% CI, 1.030–1.095; *p* < 0.001; Model 3: OR, 1.056, 95% CI, 1.025–1.088, *p* < 0.001; Model 4: OR, 1.053, 95% CI, 1.016–1.092, *p* < 0.01) and 24 h recall (Model 1: OR, 1.014; 95% CI, 1.005–1.023; *p* < 0.01; Model 2: OR, 1.019; 95% CI, 1.009–1.028; *p* < 0.001; Model 3: OR, 1.017; 95% CI, 1.008–1.027, *p* < 0.001; Model4: OR, 1.018; 95% CI, 1.007–1.028, *p* < 0.01) was found to be significantly associated with higher prevalence of subjective bad health status. Furthermore, the soft-drink group (Model 2: OR, 1.600; 95% CI, 1.160–2.208, *p* < 0.01; Model 3: OR, 1.507; 95% CI, 1.090–2.084, *p* < 0.05; Model 4: OR, 1.656; 95% CI, 1.201–2.282, *p* < 0.01) was significantly associated with higher prevalence of subjective bad health status compared with the non-soft-drink group.

With respect to depression, for Model 1, the soft-drink intake in daily increments of 10 mL according to the FFQ (OR, 1.066; 95% CI, 1.032–1.100; *p* < 0.001) and the 24 h recall (OR, 1.014; 95% CI, 1.000–1.028; *p* < 0.05) was found to be significantly associated with a higher prevalence of depression. After the adjustments, only the soft-drink intake in daily increments of 10 mL according to the FFQ (Model 2: OR, 1.058; 95% CI, 1.022–1.095; *p* < 0.01; Model 3: OR, 1.053, 95% CI, 1.019–1.089, *p* < 0.01) was found to be significantly associated with higher prevalence of depression.

## 4. Discussion

This study aimed to identify the association between soft-drink intake and obesity, depression, and subjective health status among men and women by investigating their obesity, depression status, and subjective health status based on the 2016 KNHANES VII-1. The relationship between soft-drink intake and obesity was investigated, and the findings of this study indicated that the soft-drink group was significantly associated with a higher risk of obesity, poor subjective health status, and depression in women (OR, 1.658; 95% CI, 1.237–2.223; *p* < 0.001) and, in contrast, with a lower risk of depression in men compared with the non-soft-drink group. 

In addition, the higher daily intake of soft drinks has been shown to be linked to an increased risk of obesity; the risk increased 1.040 times (95% CI, 1.016–1.066; *p* < 0.01) and 1.013 times (95% CI, 1.004–1.022; *p* < 0.01), respectively, in increments of 10 mL in daily soft-drink intake according to FFQ and 24 h recall. However, while different findings for soft-drink intake were obtained by 24 h recall (0 mL–1659 mL) and FFQ (3.3 mL–600 mL), the underweight ratio was higher in the soft-drink group than in the non-soft-drink group. Although we did not analyze soft-drink intake according to BMI, it was considered that the soft-drink intake may have been lower among underweight women, even if they drank soft drinks.

Several studies have reported that carbonated drink intake increases the risk of obesity. The more frequent consumption of sugar-sweetened soft drinks at baseline was associated with a higher waist circumference and a modest increase in the overall risks of obesity-related cancers [31]. High consumption of caloric soft drinks was associated with a high total body fat percentage (TBF%) among men in Mexico, and this may be predictive of the future development of metabolic abnormalities [32]. Malik et al. suggested that the likely mechanism via which sugar-sweetened beverages can lead to weight gain is the low satiety of liquid carbohydrates and the resulting incomplete energy compensation at subsequent meals [4]. It has also been suggested that the high contents of fructose in the corn syrup of soda may have a particular role in the risk of adiposity and diabetes [33]. 

The relationship between soft-drink intake and subjective health status and depression was also investigated. There were no differences in the subjective health status or prevalence of depression in women in the soft-drink group. However, the soft-drink group (OR, 1.600; 95% CI, 1.160–2.208; *p* < 0.01) and 10 mL increments of daily soft-drink intake according to 24 h recall (OR, 1.019; 95% CI, 1.009–1.028; *p* < 0.001) or FFQ (OR, 1.062; 95% CI, 1.030–1.095; *p* < 0.001) were associated with an increased risk of poor subjective health status or depression when the variables of age, education levels, marital status, and drinking were adjusted. An independent relationship between increased soft-drink intake and the risk of obesity and subjective poor health status in women (Models 3 and 4) was observed. However, the risk of depression with soft drinks for both women and men disappeared when the subjective health status was adjusted (Model 4). Therefore, soft drinks were shown to have an indirect effect on the subjective health status. While few studies have examined the relationship between soft-drink intake and subjective health status, the association between self-rated health and future depressive status has remained strong even after adjusting variables of age, sex, multi-morbidities, and baseline depression status or severities [17]. Subjective health status and depression are the key factors that influence the health-related quality of life of healthy adults and those with disease, as well as older adults with physical and mental weakness [34]. This study confirmed that soft-drink intake was independently linked to women’s’ subjective health status and, hence, related to the depression resulting from the subjective health status. 

For women, there were no differences in subjective health status or the prevalence of depression in the soft-drink intake group. However, when the related factors such as age, education levels, marital status, and drinking were adjusted, it was shown that a higher frequency of soft-drink intake, daily soft-drink intake, or usual intake of soft drinks correlated with worse subjective health status or depression. These results are consistent with those of previous studies. Chad et al. reported that a greater consumption of high-calorie sweet foods, such as chocolate, sodas, and sweetened fruit drinks, was associated with depressive symptoms [35]. In a prospective study of older adults in the United States, soft-drink consumption of 4 cans/day versus no consumption was associated with a 30% higher risk of self-reported, physician-diagnosed depression [11]. Although there have been no prospective studies in Asia, a cross-sectional study in China reported that adults who consumed soft drinks (4 cups/week) had a twofold higher prevalence of depressive symptoms than those who consumed <1 cup/week [36]. These studies indicated that higher consumption of soft drinks may have increased the risk of depressive symptoms. Soft drinks contain substantial amounts of sugar, which has been found to be associated with higher prevalence of depression and anxiety [35]. It will, thus, be important to investigate the relationship between soft drink consumption and mental symptoms and to suggest preventive strategies [11]. Sugar is also thought to have an impact on oxidative stress and to trigger inflammatory processes, both of which are linked to depression [37]. Likewise, heightened consumption of sugar contributes to the development of insulin resistance [38], which is also associated with elevated depressive symptoms [39].

An association between worse self-rated health and poor depression outcomes 6 years later was noted in a study of 585 people with major depression based on a population-based longitudinal health survey [10,40]. On the other hand, in this study, for men, there were no differences in obesity, subjective health status, or prevalence of depression in the soft drink group. Nonetheless, a higher daily soft-drink intake was associated with a lower risk of depression, with the risk decreasing 0.950 times (95% CI, 0.918–0.996, *p* < 0.05) in 10 mL increments of daily soft-drink intake according to the 24 h recall. In line with previous data from Australia, the US, and the UK, sugary drink consumption was more prevalent among men, young adults, those living in socioeconomically disadvantaged areas, and those with lower education levels [40,41,42,43].

However, for men in this study, the soft-drink group, compared with the non-soft-drink group, had a higher rate of unemployment, young people in their 20s, and higher education levels; however, there were no differences in household income levels. On the other hand, the exercise ratio in particular in the non-soft-drink group was higher than in the soft-drink group. A recent Cochrane review and meta-analysis of 35 randomized controlled trials found that exercise was moderately effective in reducing depressive symptoms relative to a control condition in depressed adults [44]. Sweet taste has been shown to alter the β-endorphin levels in animal studies [45]. Soft drinks are consumed to quench thirst after exercise; however, weight might be managed by consuming excess calories through exercise. Exercise, as well as soft drinks, is thought to be involved in a reduction in depression through relaxation. Further studies are needed on the effects of soda and exercise on depression in adults by sex.

Studies have shown that soft drink intake results in poor nutritional status. It was reported that sugar intake from soft drinks among Korean adolescents was 11.8 g, contributing 20.5% of the sugar intake from processed foods [46]. Soft drinks with high sugar content were usually consumed with meals or snacks during TV watching, which increases the risk of obesity due to the high energy density and low quality of meals [40,42]. Koreans consumed 73.6 g of sugar per capita in 2016, and carbonated beverages were the biggest source of sugar intake for adults aged 19 to 49 years, in contrast to coffee for those aged 50 to 65 years [47]. In this study, high-calorie and high-fat eating patterns were observed in the soft-drink group. Despite the high intake of simple sugars from soft drinks, the intake of carbohydrates and dietary fiber was low, and the density of vitamins and minerals such as Ca, P, Fe, K, vitamin B1, vitamin B2, niacin, and vitamin C was also low. The persistence of this nutritional imbalance can adversely affect the health status; therefore, nutrition education and interventions are required to ensure a healthy beverage intake and a balanced diet.

This study showed that soft-drink consumption resulted in a decrease in nutritional status and was associated particularly with increased risks of obesity, poor subjective health status, and depression in women, in contrast to a reduced risk of depression in men. Thus, the association of soft-drink intake with obesity, subjective health status, and depression worked in a reverse way according to sex, and there were also differences in the influence of various dietary and environmental factors between men and women; therefore, it will be necessary to establish appropriate approaches and strategies for health management according to sex.

There were several limitations to this study. Since the KNHANES used in this study was a cross-sectional study, it was difficult to explain cause-and-effect relationships. In addition, although a wide range of sweetened beverages were given in relation to sugar intake, there was a limit to the consideration of the relevance of sugar intake only by including soft-drink intake. However, large-scale data with a large number of study participants and good representativeness were used. Despite the limitations, this study is valuable in that it helps the basic understanding of the relationship between the frequency of soft-drink intake and daily soft-drink intake according to the 24 h recall or FFQ and obesity, subjective health status, and depression in male and female adults. It is recommended that, in the future, a more in-depth study should be performed on the relationship between sweetened beverage intake patterns and exercise in relation to obesity, subjective health status, and depression in male and female adults.

## 5. Conclusions

In summary, soft-drink consumption is associated with an increased risk of obesity, poor subjective health status, and depression in Korean women and significantly associated with a lower risk of depression in Korean men. Total energy intake and nutrient density of fat were significantly higher and nutrient density of carbohydrate, total dietary fiber, and micronutrients were lower in the soft-drink group than in the non-soft-drink group. Therefore, less frequent intake of soft drinks is suggested to prevent obesity and depression in Korean women. In addition, it will be necessary to establish appropriate approaches and strategies for health management according to sex.

## Figures and Tables

**Table 1 ijerph-18-10415-t001:** Sociodemographic characteristics and soft-drink intake by participants according to soft-drink intake frequency by sex.

Characteristics		Men (*n* = 1186)			Women (*n* = 1900)		
		Non-Soft Drinks (*n* = 502)	Soft Drinks (*n* = 684)	*p*-Value ^4^	Non-Soft Drinks (*n* = 1117)	Soft Drinks (*n* = 783)	*p*-Value ^4^
Age (yrs)		48.4 ± 0.6 ^1^	35.6 ± 0.5	0.001	47.0 ± 0.4	35.1 ± 0.6	0.001
	20 s	18 (6.5) ^2^	183 (34.3)	0.001	69 (8.8)	217 (37.7)	0.001
	30 s	62 (12.9)	225 (30.1)		200 (16.4)	271 (27.5)	
	40 s	129 (28.0)	176 (22.9)		308 (27.5)	178 (22.1)	
	50 s	181 (40.9)	77 (11.3)		363 (35.4)	86 (10.2)	
	60 s	112 (11.8)	23 (1.3)		177 (11.9)	31 (2.5)	
Education	≤Elementary	106 (16.0)	44 (5.4)	0.001	260 (21.7)	77 (8.6)	0.001
	≤Middle school	159 (34.4)	256 (39.5)		426 (40.6)	299 (40.4)	
	≥High school	237 (49.6)	384 (55.1)		431 (37.7)	407 (50.9)	
Marriage	No	439 (82.6)	432 (55.0)	0.001	1021 (89.0)	545 (61.3)	0.001
	Yes	63 (17.4)	252 (45.0)		96 (11.0)	238 (38.7)	
Occupation	No	78 (16.0)	130 (22.2)	0.029	465 (42.4)	359 (45.6)	0.233
	Yes	424 (84.0)	554 (77.8)		652 (57.6)	424 (54.4)	
Place of living	Urban	407 (87.1)	576 (88.5)	0.496	923 (87.1)	659 (88.8)	0.356
	Rural	95 (12.9)	108 (11.5)		194 (12.9)	124 (11.2)	
Household income	Low	39 (7.7)	58 (10.4)	0.190	99 (8.4)	66 (9.4)	0.623
	Middle–low	114 (21.4)	161 (23.5)		263 (22.2)	191 (23.0)	
	Middle–high	151 (30.5)	228 (31.4)		345 (31.9)	265 (33.7)	
	High	198 (40.3)	237 (34.7)		410 (37.4)	261 (33.8)	
Alcohol intake	No	123 (24.6)	171 (23.7)	0.788	622 (53.9)	348 (41.9)	0.001
	Yes	379 (75.5)	513 (76.3)		495 (46.1)	435 (58.1)	
Smoking status	No	311 (60.1)	411 (61.0)	0.762	1056 (93.2)	741 (94.2)	0.489
	Yes	191 (39.9)	273 (39.0)		61 (6.8)	42 (5.8)	
Physical activity	No	268 (49.7)	310 (42.6)	0.028	603 (51.9)	420 (51.3)	0.814
	Yes	234 (50.3)	374 (57.4)		514 (48.1)	363 (48.7)	
Body mass index (kg/m^2^)		24.8 ± 0.2 ^1^	24.7 ± 0.2	0.416	23.3 ± 0.1	22.9 ± 0.2	0.070
Waist–height ratio		0.51 ± 0.00	0.50 ± 0.00	0.001	0.49 ± 0.00	0.48 ± 0.00	0.001
Relative fat mass		24.2 ± 0.2	23.1 ± 0.2	0.001	35.0 ± 0.2	33.5 ± 0.3	0.001
Soft-drink intake frequency(times)	Never or seldom	502 (100.0)	-		1117 (100.0)	-	
	Once a month	-	134 (17.6)		-	209 (22.7)	
	Two to three a month	-	151 (21.4)		-	210 (27.9)	
	Once a week	-	145 (22.8)		-	179 (22.9)	
	Two to four a week	-	177 (26.4)		-	142 (19.8)	
	More than five a week	-	77 (11.6)		-	43 (6.8)	
Weekly soft-drink intake frequency (times/week)		0.0 ± 0.0	2.0 ± 0.1 (0.2, 21.0)	0.001	0.0 ± 0.0	1.5 ± 0.1 (0.2, 21.0)	0.001
Soft-drink intake by FFQ (mL/day)		0.0 ± 0.0	71.3 ± 3.5 (3.3, 900) ^3^	0.001	0.0 ± 0.0	46.5 ± 3.1 (3.3, 600)	0.001
Drinking of soft drinks by 24 h recall	No	458 (91.1)	471 (67.5)	0.001	1023 (91.7)	570 (71.7)	0.001
	Yes	44 (8.9)	213 (32.5)		94 (8.3)	213 (28.3)	
Soft-drink intake by 24 h recall (mL/day)		15.6 ± 3.7 (0, 731.4) ^3^	117.4 ± 9.7 (0, 1545.0)	0.001	10.2 ± 2.1 (0, 521.3)	81.5 ± 7.2 (0, 1659.0)	0.001

^1^ Values are expressed as means ± standard error. ^2^ Values are expressed as numbers (%). ^3^ Values are expressed as (min, max). ^4^ The *p*-value was estimated using the chi-square test and *t*-test in complex sample survey data analysis.

**Table 2 ijerph-18-10415-t002:** Health-related outcomes of participants according to frequency of soft-drink intake by sex.

Variables		Men (*n* = 1186)			Women (*n* = 1900)		
		Non-Soft Drinks (*n* = 502)	Soft Drinks (*n* = 684)	*p*-Value ^2^	Non-Soft Drinks (*n* = 1117)	Soft Drinks (*n* = 783)	*p*-Value ^2^
Obesity	Underweight (BMI 18.5 kg/m^2^)	13 (3.2) ^1^	15 (2.5)	0.768	38 (3.8)	65 (8.6)	0.001
	Normal (18.5 ≤ BMI 25 kg/m^2^)	283 (53.8)	366 (54.9)		780 (71.0)	508 (66.5)	
	Obese (BMI ≥ 25 kg/m^2^)	206 (43.0)	303 (42.6)		299 (25.2)	210 (24.9)	
Subjective health status	Good	430 (85.7)	600 (88.5)	0.200	927 (84.3)	635 (81.9)	0.186
	Bad	72 (14.3)	84 (11.5)		190 (15.7)	148 (18.1)	
Depression	No	482 (95.5)	661 (97.0)	0.222	1055 (94.5)	735 (94.0)	0.646
	Yes	20 (4.5)	23 (3.0)		62 (5.5)	48 (6.0)	

^1^ Values are expressed as numbers (%). ^2^ The *p*-value was estimated using the chi-square test.

**Table 3 ijerph-18-10415-t003:** Participants’ nutrient intake according to frequency of soft-drink intake by sex.

Variables	Men (*n* = 1186)			Women (*n* = 1900)		
	Non-Soft Drinks (*n* = 502)	Soft Drinks (*n* = 684)	*p*-Value ^2^	Non-Soft Drinks (*n* = 1117)	Soft Drinks (*n* = 783)	*p*-Value
Total energy (kcal)	2368.4 ± 49.0 ^1^	2583.4 ± 37.3	0.001	1706.0 ± 24.8	1834.5 ± 29.7	0.001
% energy of carbohydrate	57.7 ± 0.7	54.6 ± 0.6	0.001	63.8 ± 0.4	59.1 ± 0.5	0.001
% energy of protein	13.8 ± 0.2	14.8 ± 0.2	0.001	14.3 ± 0.2	14.4 ± 0.2	0.606
% energy of fat	19.4 ± 0.4	23.9 ± 0.4	0.001	20.2 ± 0.3	23.8 ± 0.4	0.001
Protein (g/1000 kcal)	34.6 ± 0.5	37.0 ± 0.5	0.001	35.7 ± 0.4	36.0 ± 0.5	0.606
Fat (g/1000 kcal)	21.6 ± 0.5	26.5 ± 0.5	0.001	22.4 ± 0.3	26.4 ± 0.5	0.001
Carbohydrate (g/1000 kcal)	144.4 ± 1.7	136.5 ± 1.4	0.001	159.5 ± 1.1	147.8 ± 1.3	0.001
Total dietary fiber (g/1000 kcal)	11.6 ± 0.2	9.3 ± 0.2	0.001	13.6 ± 0.2	10.6	0.001
Calcium (mg/1000 kcal)	242.1 ± 6.4	215.0 ± 4.5	0.001	268.6 ± 4.2	237.1 ± 4.6	0.001
Phosphorus (mg/1000 kcal)	524.4 ± 7.5	509.3 ± 5.3	0.097	572.4 ± 5.3	531.9 ± 6.1	0.001
Iron (mg/1000 kcal)	8.8 ± 0.3	7.9 ± 0.2	0.025	9.4 ± 0.2	8.1 ± 0.2	0.001
Sodium (mg/1000 kcal)	1969.8 ± 45.7	1940.2 ± 33.1	0.607	1846.7 ± 33.3	1842.2 ± 37.5	0.923
Potassium (mg/1000 kcal)	1477.1 ± 26.7	1299.0 ± 21.5	0.001	1734.6 ± 24.7	1457.1 ± 26.8	0.001
Vitamin A(ugRE/1000 kcal)	350.5 ± 17.5	313.9 ± 11.8	0.074	393.3 ± 17.1	416.2 ± 42.1	0.606
Thiamin (mg/1000 kcal)	0.996 ± 0.016	0.942 ± 0.013	0.013	1.030 ± 0.010	0.967 ± 0.014	0.001
Riboflavin (mg/1000 kcal)	0.7 ± 0.01	0.7 ± 0.01	0.285	0.8 ± 0.01	0.7 ± 0.01	0.001
Niacin (mg/1000 kcal)	8.0 ± 0.2	8.2 ± 0.1	0.446	8.7 ± 0.1	8.3 ± 0.1	0.001
Vitamin C (mg/1000 kcal)	49.0 ± 2.5	34.7 ± 1.7	0.001	77.5 ± 3.1	50.3 ± 2.7	0.001

^1^ Values are expressed as means ± standard error. ^2^ The *p*-value was estimated using the *t*-test in complex sample survey data analysis.

**Table 4 ijerph-18-10415-t004:** ORs for health-related outcomes by soft drink consumption.

Variables				Men				Women	
		Non-Soft Drinks	Soft Drinks	Continuous Variables, per Daily 10 mL Increment (FFQ)	Continuous Variables, per Daily 10 mL Increment (24 h Recall)	Non-Soft Drinks	Soft Drinks	Continuous Variables, per Daily 10 mL Increment (FFQ)	Continuous Variables, per Daily 10 mL Increment (24 h Recall)
Obesity (BMI 25)	Model 1 ^2^	1	0.983 (0.788–1.227) ^1^	1.003 (0.988–1.018)	1.000 (0.993–1.007)	1	0.984 (0.760–1.275)	1.006 (0.980–1.033)	1.003 (0.994–1.012)
	Model 2 ^3^	1	0.910 (0.699–1.186)	1.002 (0.985–1.019)	0.999 (0.991–1.007)	1	1.658 (1.237–2.223) ***	1.040 (1.016–1.066) **	1.013 (1.004–1.022) **
	Model 3 ^4^	1	0.941 (0.824–1.076)	0.998 (0.990–1.007)	1.000 (0.996–1.004)		1.658 (1.237–2.221) ***	1.036 (1.010–1.063) **	1.012 (1.003–1.021) **
	Model4 ^5^	1	0.912 (0.700–1.187)	1.001 (0.985–1.018)	0.999 (0.992–1.007)		1.593 (1.183–2.146) **	1.034 (1.010–1.058) **	1.011 (1.002–1.020) *
Subjective health status (bad)	Model 1	1	0.775 (0.524–1.148)	1.005 (0.984–1.026)	0.990 (0.979–1.002)	1	1.181 (0.920–1.515)	1.042 (1.016–1.070) **	1.014 (1.005–1.023) **
	Model 2	1	1.013 (0.643–1.596)	1.012 (0.991–1.033)	0.993 (0.979–1.007)	1	1.600 (1.160–2.208) **	1.062 (1.030–1.095) ***	1.019 (1.009–1.028 ***
	Model 3 ^6^	1	1.032 (0.656–1.627)	1.012 (0.992–1.033)	0.993 (0.979–1.007)	1	1.507 (1.090–2.084) *	1.056 (1.025–1.088) ***	1.017 (1.008–1.027) ***
	Model 4 ^4^	1	1.106 (0.685–1.787)	1.016 (0.994–1.038)	0.997 (0.983–1.010)	1	1.656 (1.201–2.282) **	1.053 (1.016–1.092) **	1.018 (1.007–1.028) **
Depression (PHQ 10)	Model 1	1	0.655 (0.329–1.304)	0.982 (0.934–1.031)	0.957 (0.918–0.998) *	1	1.102 (0.726–1.672)	1.066 (1.032–1.100) ***	1.014 (1.000–1.028) *
	Model 2	1	0.577 (0.230–1.445)	0.969 (0.907–1.037)	0.950 (0.906–0.996) *	1	0.926 (0.533–1.610)	1.058 (1.022–1.095) **	1.011 (0.998–1.025)
	Model 3 ^6^	1	0.643 (0.276–1.494)	0.975 (0.917–1.036)	0.953 (0.911–0.997) *	1	0.867 (0.495–1.519)	1.053 (1.019–1.089) **	1.010 (0.997–1.023)
	Model 4 ^5^	1	0.580 (0.201–1.675)	0.962 (0.893–1.036)	0.947 (0.895–1.003)	1	0.757 (0.449–1.276)	1.030 (0.991–1.070)	1.003 (0.989–1.016)

^1^ Values are expressed as odds ratios (confidence intervals). ^2^ Model 1: crude. ^3^ Model 2: adjusted for age, education, marital status, occupation, physical activity, and total energy intake in men; adjusted for age, education, marriage, alcohol drinking, and total energy intake in women.^4^ Model 3 or Model 4: Model 2 + additionally adjusted for depression. ^5^ Model 4: Model 2 + additionally adjusted for subjective health status. ^6^ Model 3: Model 2 + additionally adjusted for obesity. * *p* 0.05, ** *p* 0.01, *** *p* 0.001.

## Data Availability

All the raw data of survey is available http://knhanes.cdc.go.kr/ (accessed on 2 October 2021). The data presented in this study are available on reasonable request from the corresponding author.

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
