# Peer review of "Association between Soft-Drink Intake and Obesity, Depression, and Subjective Health Status of Male and Female Adults"

_ijerph, 2021, doi:10.3390/ijerph181910415_

Round 1
Reviewer 1 Report
Review of
The association between soft drinks intake and obesity, depression and the subjective health statuses of adult men and women
- Title: please change statuses to Also, throughout the manuscript as it occurs frequently.
- Abstract: were the calorie and nutrients significantly higher in the soft drink group?
- The goal does not seen appropriate since this is not an intervention study.
- 1. Ethics approval number.
- 2. How was physical activity classified?
- 3.1. Reference required for BMI and cur-off points.
- Reference 23 is incorrect. It is for food composition tables and not for the KNHANES
- 3.1. Describe how measurements were done and provide scale and stadiometer details. Were scales calibrated? BMI >= 25 is usually described as overweight and obese
- 4.1. Describe the method of the 24-hour recall and how portions sizes were determined.
- 4.2. How was the FFQ validated? What is meant by semi-quantitative?
- The portion sizes 0.5 and 1.0 should not have %.
- Avoid brand names but define soft drinks
- This sentence is not clear: Second, the soft drink intakes, including that of Coke, Sprite, and other soft drinks beverages, were analyzed using a 24-h dietary recall. Was it based on portion size per 24 hours?
- 5. Why were the groups divided into < or ≥ one soft drink per month. This means that someone who only had soft drinks once or twice a month are included as regular users. What is the motivation for this? I would imagine using one or more per week would be better. It would also comply with the literature.
- 5 occupation not occupations and level of physical activity not physical activities.
- Table 1. The 1 and 2 are the wrong way round. I would suggest doing the table in landscape
- Table 2- although the category is significant for women soft drink users and non-soft drink used were 25% who were obese although less were in the Normal BMI category. No means are given in Table 2, so remove the superscript under the table.
- P values in the text should not be capitals
- Discussion . Needs a serious language check.
Author Response
We appreciate your helpful comments on this manuscript. We tried to address the issues the reviewer pointed out or suggested in the comments as much as we could and the revisions made were highlighted in red in the Word file of text. Also, we provided answers to the reviewer’s comments as follows:
1. Title: please change statuses to Also, throughout the manuscript as it occurs frequently.
-> (Answer) We changed statuses to status throughout the manuscript as suggested.
2. Abstract: were the calorie and nutrients significantly higher in the soft drink group?
-> (Answer) We added the statistical significance of nutritional results in abstract.
3. The goal does not seem appropriate since this is not an intervention study.
-> (Answer) That inappropriate sentence was deleted as suggested.
4. 1.1.Ethics approval number.
-> (Answer) The KNHANES surveys (http://knhanes.cdc.go.kr/) were approved by the Institutional Review Board of the Korean Center for Disease Control and Prevention (KCDC) (2010- 02CON-21-C, 2011-02CON-06-C, 2012-01EXP-01-2C, 2013-07CON-03-4C, 2013- 122EXP-03-5C). However, the 2016 KNHANES were exempted from research ethics approval based on the Bioethics and Safety Act. So, we described that “The 2016 KNHANES was exempted from the research ethics approval from the Bioethics and Safety Act of the Institutional Review Board of the Korea Centers for Disease Control and Prevention. And all participants provided written informed consent.” (page 2)
5. 2. How was physical activity classified?
->(Answer) As suggested, we described that “Physical activity was also divided into two categories: (practice physical activity of mid-intensity over 2 1/2hrs. per week or of high-intensity or mid-and high mixed intensity over 1 hr. 15 min per week (1 min. of high-intensity physical activity equals 2 min. of mid-intensity physical activity) and no (no exercise) according to the KNHANES’s guideline by the KCDC [21, 22].].” (page 3)
6. 3.1. Reference required for BMI and cur-off points.
->(Answer) As suggested, we described the reference for BMI and cut-off points. “Body mass index (BMI) was calculated as weight (kg) divided by height squared (m2) and the participants were classified into three categories based on their BMIs (kg/m2): underweight (BMI < 18.5), normal weight (18.5 ≤ BMI < 25), or obesity (BMI ≥ 25) according to the KNHANES‘ guideline by the KCDC [21].” (page 3)
7. Reference 23 is incorrect. It is for food composition tables and not for the KNHANES
->(Answer) As pointed out, we corrected the reference. (page 3)
8. 3.1. Describe how measurements were done and provide scale and stadiometer details. Were scales calibrated? BMI >= 25 is usually described as overweight and obese
->(Answer) The following details were added as suggested.
“Height and waist were measured to the nearest 0.1 cm. Similarly, weight was measured to the nearest 0.1 kg in light clothing, with shoes off, using a calibrated metric weight scale.” (page 2)
9. 4.1. Describe the method of the 24-hour recall and how portions sizes were determined.
->(Answer) We cited the method of the 24-h recall and portion sizes in reference: “Dietary intake data were obtained using a one-day 24-h dietary recall protocol as part of the nutrition survey [27].” The portions were categorized as one of the three sizes depending on the type of food consumed: 0.5, 1.0, and 1.5 portions as presented in the KNHANES‘ guideline by the KCDC [20, 30]. (page 3)
10. 4.2. How was the FFQ validated? What is meant by semi-quantitative?
->(Answer) We cited FFQ validation in reference and corrected semi-quantitative into quantitative. “Soft drink intake data were obtained using two dietary intake survey methods: (1) validated quantitative food frequency questionnaire (FFQ) [29] and (2) 24-h dietary recall.”
11. The portion sizes 0.5 and 1.0 should not have %.
->(Answer) % was deleted as suggested
12. Avoid brand names but define soft drinks
->(Answer) We defined soft drinks, using generic terms. “Soft drink” intake was defined as “carbonated drinks” (i.e. coke, cider, and other soft drink beverages) in the questionnaire in this study. (page 3)
13. This sentence is not clear: Second, the soft drink intakes, including that of Coke, Sprite, and other soft drinks beverages, were analyzed using a 24-h dietary recall. Was it based on portion size per 24 hours?
->(Answer) As suggested, we re-wrote this sentence clearly.
“Second, the soft drink intake by 24-h dietary recall was analyzed by summing and calculating per day consumption (mL) of soft drinks including coke, cider, and other soft drink beverages in accordance with soft drink classification of FFQ.”
14. Why were the groups divided into < or ≥ one soft drink per month. This means that someone who only had soft drinks once or twice a month are included as regular users. What is the motivation for this? I would imagine using one or more per week would be better. It would also comply with the literature.
->(Answer) We agree with the reviewers that grouping by per week is more appropriate. But we attempted to examine the effect of those who consumed soft drinks at least once compared to those who did not consume soft drink at all. In this study, we tried to classify the group into two groups, the group who did not drink soda at all and the group who drank soda at least once. In the 9 categories of soft dink intake frequency surveyed by the FFQ, the response of not consuming soft drink at all was designated as the non-soft drink group (“never or seldom”), and the response of consuming it at other frequencies was classified as the soft drink intake group (≥ once a month) (including “ once a month”, “two to three times a month”, “one to two times a week”, “three to four times a week”, “five to six times a week”, “once a day”, “twice a day” and “three times or more every day”). In order to eliminate confusion about the frequency of intake when understanding the results, the frequency of intake was deleted from the group name, and the manuscript was also revised.
15. 5 occupation not occupations and level of physical activity not physical activities.
-> (Answer) Corrected as suggested
16. Table 1. The 1 and 2 are the wrong way round. I would suggest doing the table in landscape
-> (Answer) Corrected as suggested
17. Table 2- although the category is significant for women soft drink users and non-soft drink users were 25% who were obese although less were in the Normal BMI category. No means are given in Table 2, so remove the superscript under the table.
->(Answer) We deleted the superscript of means.
18. P values in the text should not be capitals.
-> (Answer) Corrected as suggested
19. Discussion . Needs a serious language check.
-> (Answer). We had our English manuscript proofread and edited at Editage (www.editage.co.kr), a professional English translation service.
Reviewer 2 Report
In my opinion abstract of the work is correctly written and contains all the most important information. The introduction of the work should be editorially checked, as it contains linguistic and editorial errors.
Inclusion and exclusion criteria are correct, but note = the BMI is not the accurate markers of body composition. In the study, you should use other parameters to assess body composition e.g. body composition test (with % of total body fat, muscle mass), WHR, WHtR or RFM markers.
Moreover, there is a large disproportion between the group of women and men participating in the study.
Why did the authors of the study only describe the meals consumed in the last 24 hours? It is a very unreliable measurement that describes eating habits in a slight way.
The results are described in a very comprehensive but correct way.
The conclusions of the work are presented correctly, however, the critique of the methodology should be placed at the end of the discussion, not in the conclusions.
Author Response
We appreciate your helpful comments on this manuscript. We tried to address the issues the reviewer pointed out or suggested in the comments as much as we could and the revisions made were highlighted in red in the Word file of text. Also, we provided answers to the reviewer’s comments as follows:
- In my opinion abstract of the work is correctly written and contains all the most important information. The introduction of the work should be editorially checked, as it contains linguistic and editorial errors.
(Answer) We checked and modified the linguistic and editorial errors in the introduction. We had our English manuscript proofread and edited at Editage, a professional English translation service. (www.editage.co.kr). - Inclusion and exclusion criteria are correct, but note = the BMI is not the accurate markers of body composition. In the study, you should use other parameters to assess body composition e.g. body composition test (with % of total body fat, muscle mass), WHR, WHtR or RFM markers.
(Answer) As suggested, we added the results of body composition parameters of WHtR and RFM in table 1 (Page 5, 6). It would be meaningful to conduct further research using the new body composition parameters you suggested.
- Moreover, there is a large disproportion between the group of women and men participating in the study.
(Answer) As pointed out, there is a disproportion between the group of women and men. Also, there is a difference in the dietary behaviors such as soft drink intake and its effects according to gender. Therefore, in order to exclude these gender-dependent effects, confounding factors were adjusted for male and female respectively.
- Why did the authors of the study only describe the meals consumed in the last 24 hours? It is a very unreliable measurement that describes eating habits in a slight way.
(Answer) As pointed out, I agree that the dietary intake survey by the 24-h recall method has limitations in reflecting the usual intake. However, as a result of analyzing the relationship between soft drink intake and health-related outcomes in this study, the association between soft drink intake by the FFQ survey as well as by the 24-h recall method was found. Also, soft drink intake trends estimated by FFQ and 24-h recall were similar. In addition, the 24-hour recall method can more accurately identify the soft drinks consumed for one day. Therefore, we analyzed nutritional status using the 24-h recall dietary data in this study.
5.The results are described in a very comprehensive but correct way. The conclusions of the work are presented correctly, however, the critique of the methodology should be placed at the end of the discussion, not in the conclusions.
(Answer) We placed the limitation of this study at the end of the discussion.
Round 2
Reviewer 1 Report
The following still need to be addressed.
8.3.1. Make of scale and anthropometer eg Seca 104.
Weighing method and measuring methods need to be described in detail. Not just a reference
Calibration method has not been described. How often was it done and was it tested against a standard weight?
9.4.1 The 24-h needs to be described in detail eg. The person reported all food items consumed over the previous 24 hours? multiple pass method followed ? were food models shown?
Please give a brief description of the validation of the FFQ and do not just refer to a reference.
12.&13. Coke is a brand name and should be removed.
Author Response
We appreciate your helpful comments on this manuscript. We tried to address the issues the reviewer pointed out or suggested in the comments as much as we can and the revisions made were highlighted in red in the Word file of text. Also, we provided answers to the reviewer’s comments as follows:
- The following still need to be addressed. 8.3.1. Make of scale and anthropometer eg Seca 104. Weighing method and measuring methods need to be described in detail. Not just a reference Calibration method has not been described. How often was it done and was it tested against a standard weight?
(Answer) The following details were added as suggested.
“Anthropometric data were collected by a team of experts by using a consistent, standardized methodology during the health examinations. Height was measured to the nearest 0.1 cm using stadiometer (Seca 225, seca., Germany). Waist circumference was measured using a measuring tape (Seca 200; Seca) to the nearest 0.1cm. Similiarly, weight was measured to the nearest 0.1 kg in light clothing, with shoes off, using a calibrated balance scale (GL-6000-20, G-tech, Korea). To maintain the accuracy of the measurements, the instruments have been replaced with newly calibrated devices every year.”(page 2)
- 9.4.1 The 24-h needs to be described in detail eg. The person reported all food items consumed over the previous 24 hours? multiple pass method followed ? were food models shown?
(Answer) The following details were added as suggested.
“Detailed descriptions of the type and amount of all foods consumed on the day before the interview, including snacks, beverages, and ingredient information for all sauces and condiments were requested. Additionally, information regarding the time and the place of each meal was recorded. The 24-h dietary recall began with the first meal or beverage consumed at waking until midnight of the reporting day. Trained dietitians administered the 24-h dietary recall using measuring cups, portion-size booklets and photographs. All completed records were checked by a research dietitian for accuracy.”(page 3)\
- Please give a brief description of the validation of the FFQ and do not just refer to a reference.
(Answer) The following brief description were added as suggested.
“The FFQ was developed based on dietary intake data from the KNHANES and has been previously validated using dietary records taken over 3 days during each of the four seasons in 124 participants as described elsewhere [29].”(page 3)
- 12.&13. Coke is a brand name and should be removed.
(Answer) As you suggested, we deleted Coke and modified it with cola.
Also, we corrected spelling errors throughout the manuscript.
